# Fast Multipole Attention: A Divide-and-Conquer Attention Mechanism for Long Sequences

## Abstract

Transformer-based models have achieved state-of-the-art performance in many areas. However, the quadratic complexity of self-attention with respect to the input length hinders the applicability of Transformer-based models to long sequences. To address this, we present Fast Multipole Attention (FMA), a new attention mechanism that uses a divide-and-conquer strategy to reduce the time and memory complexity of attention for sequences of length $n$ from $\mathcal{O}(n^2)$ to $\mathcal{O}(n \log n)$ or $\mathcal{O}(n)$, while retaining a global receptive field. The hierarchical approach groups queries, keys, and values into $\mathcal{O}(\log n)$ levels of resolution, where groups at greater distances are increasingly larger in size and the weights to compute group quantities are learned. As such, the interaction between tokens far from each other is considered in lower resolution in an efficient hierarchical manner. The overall complexity of FMA is $\mathcal{O}(n)$ or $\mathcal{O}(n \log n)$, depending on whether the queries are down-sampled or not. This multi-level divide-and-conquer strategy is inspired by fast summation methods from $n$-body physics and the Fast Multipole Method. FMA is similar in spirit to recently proposed hierarchical attention mechanisms such as H-transformer by Zhu & Soricut and Multi-Resolution Analysis (MRA) attention by Zeng et al., but our key advance is that we *learn* the basis functions that compute group quantities. We perform evaluation on autoregressive and bidirectional language modeling tasks and compare our FMA model with other efficient attention variants on medium-size datasets. We find empirically that the Fast Multipole Transformer performs much better than other efficient transformers in terms of memory size and accuracy. The FMA mechanism has the potential to empower large language models with much greater sequence lengths, taking the full context into account in an efficient, naturally hierarchical manner during training and when generating long sequences.

## 1 Introduction

The Transformer (Vaswani et al., 2017) was originally introduced in the context of neural machine translation (Bahdanau et al., 2014) and has been widely adopted in various research areas including image recognition (Dosovitskiy et al., 2021), music generation (Huang et al., 2019), speech recognition (Gulati et al., 2020), and protein structure prediction (Jumper et al., 2021). In natural language processing, pre-trained Transformer-based language models have shown remarkable performance when combined with transfer learning (Devlin et al., 2019; Brown et al., 2020). At the heart of Transformer models is the self-attention mechanism, which essentially computes weighted averages of inputs based on similarity scores obtained with inner products. Unlike recurrent neural network layers and convolutional layers, self-attention layers have a global receptive field, allowing each attention layer to capture long-range dependencies in the sequence.

However, this advantage comes with time and space complexity of $\mathcal{O}(n^2)$ for inputs of length $n$, since self-attention computes $n$ similarity scores for each token. This quadratic cost limits the maximum context size feasible in practice. To address this problem, many methods focusing on improving the efficiency of self-attention have been proposed. One line of work is sparsification of the attention matrix with either fixed patterns (Child et al., 2019; Beltagy et al., 2020) or clustering methods (Kitaev et al., 2020; Roy et al., 2021). However, the ability of attention to capture information from the entire

sequence is impaired by these modifications. Another direction is to linearize attention either by kernelizing (Choromanski et al., 2021; Katharopoulos et al., 2020) or replacing `softmax` with a linear operation (Qin et al., 2022), and avoiding to form the attention matrix explicitly. Although these approaches have complexity linear in $n$, their main disadvantage is that in autoregressive models they require recurrent rather than parallel training along the sequence length dimension (Tay et al., 2022). Since generating or translating long texts poses document-level challenges of *discourse* such as lexical cohesion, coherence, anaphora, and deixis (Maruf et al., 2021), the ability to include more context in language models is important.

In this work, we propose an efficient variant of attention called *Fast Multipole Attention (FMA)*, which uses different resolutions according to the distance between the input and output tokens. Specifically, in the neighborhood of the query, attention is calculated with keys and values in full resolution. Moving away from the query, keys and values are grouped and downsampled (or *summarized*) in increasingly larger groups using learned downsampling weights, and attention is calculated in lower resolution. The resulting attention matrix has a hierarchical structure, as in Figure 1b. FMA preserves the global receptive field of full attention and has an overall complexity of $\mathcal{O}(n \log n)$ or $\mathcal{O}(n)$, depending on whether the queries are downsampled or not. It can be used in both autoregressive and bidirectional settings, as used, for example, in the GPT family and BERT family of language models.

The motivation behind the divide-and-conquer strategy of FMA is not just an efficient computational technique to make the memory and computation cost of large context sizes manageable; the multilevel strategy is also a natural hierarchical way to handle large context sizes that intuitively makes sense for many applications. For example, when humans read books they don't remember the context provided by each individual word on previous pages or in previous chapters. Rather, they *summarize* in their mind the context of what they have read, with more detail for recent pages than for earlier chapters. Similarly, when human authors write books, they don't keep the detailed context of every previous word in mind, but they work with a summarized context with a level of detail that, at least to some extent, can be assumed hierarchical-in-distance. FMA provides a natural hierarchical model for this summarization process that makes large context sizes manageable, both for humans and for language models. As such, FMA can be used to build a class of divide-and-conquer language models that enable large context sizes using a hierarchical context summarization mechanism.

This multi-level divide-and-conquer strategy is inspired by fast summation methods from $n$-body physics and the Fast Multipole Method (Barnes & Hut, 1986; Greengard & Rokhlin, 1987; Beatson & Greengard, 1997; Martinsson, 2015). Compared to other recently proposed and very successful hierarchical attention mechanisms such as H-transformer (Zhu & Soricut, 2021) and Multi-Resolution Analysis (MRA) attention (Zeng et al., 2022), the key advance of our work is that we *learn* the basis functions that compute group quantities. To evaluate the effectiveness of our method, we experiment on autoregressive and masked (bidirectional) language modeling tasks. We show that FMA can handle much longer sequences as it uses less memory and FLOPs. In addition, we show that our FMA outperforms other efficient attention variants in terms of efficiency and efficacy.

## 2 RELATED WORK

Reducing the asymptotic complexity of attention is crucial to scaling Transformer-based models for long inputs. In this section, we summarize relevant works on improving the efficiency of attention and broadly divide them into the following categories.

**Sparse patterns** Local attention is widely used in (Liu et al., 2018; Child et al., 2019; Ainslie et al., 2020; Beltagy et al., 2020; Zaheer et al., 2020) due to its simplicity. It calculates entries on the diagonals or block diagonals of the attention matrix and has a linear cost. Another pattern is strided attention, which computes entries of the attention matrix on dilated diagonals, as in (Child et al., 2019; Beltagy et al., 2020). Strided attention can achieve a larger receptive span than local attention. Global tokens are another commonly used pattern. These tokens summarize information from the whole sequence and can be read by all output positions. They can be tokens within the input sequence (Child et al., 2019; Beltagy et al., 2020; Zaheer et al., 2020) or auxiliary tokens (Ainslie et al., 2020; Zaheer et al., 2020).

**Clustering** These approaches group inputs into clusters and compute attention within each cluster, in order to reduce the amount of computation. They are often based on the observation that the

output of `softmax` is dominated by the largest inputs. In these works, k-means clustering (Roy et al., 2021; Wang et al., 2021) and locality sensitive hashing (Kitaev et al., 2020; Vyas et al., 2020) are commonly used. ClusterFormer (Wang et al., 2022) makes use of soft clustering by updating the word embeddings and the centroids simultaneously.

**Linearization** Linearization aims to replace or approximate `softmax` with a linear operator to avoid forming the quadratic attention matrix explicitly. Linear Transformer (Katharopoulos et al., 2020) replaces the exponentiated dot-product $\exp(q^\top k)$ by $\phi(q)^\top \phi(k)$ with $\phi(\cdot) = \texttt{elu}(\cdot) + 1$, where `elu`$(\cdot)$ denotes the exponential linear unit activation function. Performer (Choromanski et al., 2021) and Random Feature Attention (Peng et al., 2021) both use random features to define the kernel function $\phi$. cosFormer (Qin et al., 2022) replaces `softmax` with the `ReLU` function followed by a cosine re-weighting mechanism that encourages locality. A drawback of linearization methods is that during autoregressive training, they need to be run recurrently, instead of via the more efficient teacher forcing approach (Tay et al., 2022). FMMformer (Nguyen et al., 2021) presents a 2-level method where the coarse low-rank approximation uses linearization.

**Downsampling** Memory-compressed attention (Liu et al., 2018) downsamples keys and values using convolution kernels of size 3 with stride 3. Although the memory cost is reduced by a factor of 3, it scales quadratically in $n$. Linformer (Wang et al., 2020) projects keys and values from $\mathbb{R}^{n \times d}$ to $\mathbb{R}^{k \times d}$ with linear projections, resulting in an attention matrix, $QK^\top$, of size $n \times k$. However, causal masking in such downsampling methods that mix past and future is difficult.

**Hierarchical and multi-resolution attention** Our work is most closely related to the recently proposed Hierarchical matrix transformer (H-transformer) (Zhu & Soricut, 2021) and Multi-Resolution Analysis (MRA) attention (Zeng et al., 2022). Both H-transformer and MRA attention approximate self-attention with a local+global (near-field+far-field) decomposition (see also FMMformer (Nguyen et al., 2021)), and make the crucial next step to use $\mathcal{O}(\log n)$ levels of resolution recursively to obtain $\mathcal{O}(n)$ efficiency in memory and time, while retaining high approximation accuracy. H-transformer (which is related to the fast multipole method, see Section A.10 for more details) uses fixed averaging between levels and by the nature of the hierarchical matrix format places the higher resolutions closer to the diagonal of the attention matrix. The MRA attention of (Zeng et al., 2022) makes a seminal next step in *adaptively* choosing higher recursive resolution where the attention weights are the largest in the attention matrix (with higher resolution not necessarily occurring closer to the diagonal, especially, for example, for two-dimensional data such as images). Indeed, multi-resolution analysis can be seen as an *adaptive* generalization of the fast multipole method (Beylkin, 1996). Using a judiciously chosen fixed wavelet family, MRA attention provides a procedure with strong advantages across both performance/accuracy and efficiency compared to other efficient attention mechanisms and full attention (Zeng et al., 2022).

The crucial advance in our work, both relative to H-transformer (Zhu & Soricut, 2021) and MRA attention (Zeng et al., 2022), is that our FMA approach *learns* the basis functions that compute group quantities, as opposed to the fixed averaging basis functions of H-transformer (Zhu & Soricut, 2021) and the fixed wavelet family used in MRA attention (Zeng et al., 2022). Our numerical results show that *learning the basis functions* leads to substantially better accuracy for language tasks compared to H-transformer, for the same memory and time cost. While our FMA does not possess the flexibility of MRA to adaptively determine regions of high resolution in the attention matrix, we do find results for language modelling that have substantially better accuracy than the MRA attention from (Zeng et al., 2022). As such, our novel results demonstrate the key importance of *learning the basis functions* in hierarchical attention approximations. At the same time, our findings strongly suggest that future work combining our approach in FMA to learn the basis functions with the adaptivity of multi-resolution methods like the MRA attention from (Zeng et al., 2022) may lead to the next quantum leap in designing accurate, efficient and versatile multilevel transformer methods in the future.

## 3 FAST MULTIPOLE ATTENTION

### 3.1 ATTENTION MECHANISM

Throughout the paper, indices start from 0. We use subscript indices to indicate rows of matrices. More notation is recalled in the Appendix.

The Transformer is characterized by the attention mechanism, which is capable of capturing long-distance dependencies from the entire sequence. For *queries* $Q \in \mathbb{R}^{n \times d}$, *keys* $K \in \mathbb{R}^{n \times d}$ and *values* $V \in \mathbb{R}^{n \times d}$, the *(unnormalized) attention matrix* is

$$C = QK^\top, \tag{1}$$

and the *scaled dot-product attention* is defined by

$$\texttt{ATTN}(Q, K, V) = \texttt{softmax}\left(\frac{C}{\sqrt{d}}\right)V. \tag{2}$$

Here $n$ is the context size, $d$ is the embedding dimension for attention, and $\texttt{softmax}$ is applied row-wise. Essentially, the attention mechanism calculates a convex combination of the value vectors in the rows of $V$ with coefficients obtained from inner products between query and key vectors. In the case of self attention, $Q$, $K$ and $V$ are obtained from the same embedding $X \in \mathbb{R}^{n \times d_{embed}}$ with learned linear projections: $Q = W_Q X$, $K = W_K X$, and $V = W_V X$. Calculating the attention matrix according to (1) requires $\mathcal{O}(n^2 d)$ time and $\mathcal{O}(n^2)$ space and becomes a major computational bottleneck for large $n$.

### 3.2 FAST MULTIPOLE ATTENTION

Fast Multipole Attention considers the input sequence at different resolutions. The dot-product attention is evaluated directly in the neighborhood of the query. For each query, distant keys and values are divided into groups and a *summarization* is computed for each group via downsampling. The downsampled key vector is used for calculating the attention score, which later is multiplied with the downsampled value vector to obtain the attention output. By splitting the entire sequence into a tree of intervals, FMA achieves loglinear space and time complexity with respect to the sequence length and hierarchical summarization. When the queries are also downsampled, we obtain an $\mathcal{O}(n)$ version of FMA, named FMA-linear. In this section, we only describe in detail our $\mathcal{O}(n \log n)$ version of FMA, and we describe FMA-linear in the supplementary material. Figure 1a demonstrates the intuition behind FMA. The rest of this section will describe FMA in detail.

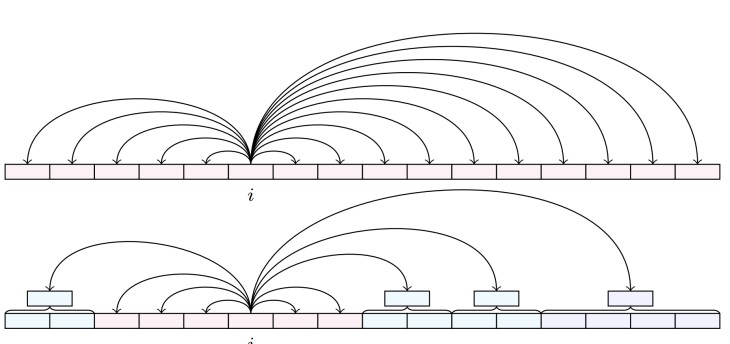
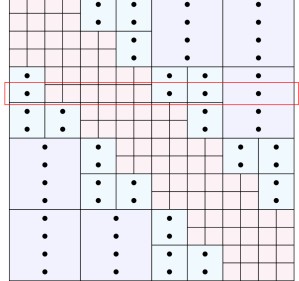

(a) Top: full attention. All pairwise interactions are computed. Bottom: FMA with 3 levels and group size $m = 2$. In the neighborhood of a token $i$, attention is computed in full. Tokens far away from $i$ are grouped and downsampled/summarized. The red, green, and purple blocks correspond to the fine level, the first coarse level, and the second coarse level, respectively.

(b) The structure of the FMA attention matrix $\widetilde{C} \approx QK^\top$ with 3 levels for group size $m = 2$ and approximation rank $p = 1$. The highlighted row corresponds to Figure 1a. Dots represent downsampled query-key products.

Figure 1: Illutration of Fast Multipole Attention (FMA). The influence of the context sequence on token $i$ is computed in a divide-and-conquer way. First, the tokens in the sequence are divided between the immediate neighborhood of $i$ and the rest of the sequence. The influence of the tokens near $i$ (level 0) is computed using full attention. Next, the remaining tokens are divided between the level-1 tokens and the further-away tokens, and the influence of the level-1 tokens is computed using level-1 summarization. This divide-and-conquer strategy is repeated for all subsequent increasingly coarser levels, resulting in an efficient algorithm with complexity $\mathcal{O}(n \log n)$.

**Tree Structure** To describe our FMA algorithm, we introduce a hierarchical tree structure for dividing the input sequence. We denote the fine-level group size by $m$, which is small compared

to $n$. On the fine level, level 0, tokens are considered individually. Level 1 contains intervals of size $m_1 := m$. The interval size on the $\ell$-th level is $m_\ell := 2^{\ell-1}m$ for $\ell = 1, \ldots, L$, where $L := \log_2(n/m) - 1$. The coarsest level, level $L$, contains intervals of size $n/4$. We assume $n$ is divisible by $m$ and $n/m$ is a power of 2.

**Downsampling Keys and Values** On level $\ell = 1, \ldots, L$, FMA downsamples each interval of size $m_\ell \times d$ in $K$ and $V$ to $p \times d$, where $p$ is the approximation rank of the off-diagonal blocks and $m$ is divisible by $p$. When $p = 1$, each group is summarized by one vector. In our numerical results, we use $p = 4$ summary vectors for each block to increase the accuracy of approximation (similar to using multiple heads in attention, or retaining multiple terms in the truncated multipole expansion for the original fast multipole method (Greengard & Rokhlin, 1987; Beatson & Greengard, 1997)). We adopt 1D convolution as the downsampling operator with `kernel_size = m_ℓ`, `stride = m_ℓ`, `groups = d`. For each level, with group size $m_\ell$, the 1D convolution learns $p$ convolution kernels of size $m_\ell$ for each of the $d$ features. This produces $p$ summarized vectors in each group of size $m_\ell$. Therefore, the downsampled key/value matrices, $K^{(\ell)}$ and $V^{(\ell)}$, have size $\frac{np}{m_\ell} \times d$.

**Hierarchical Matrix Structure** We introduce a hierarchical structure for an $n \times n$ matrix with the following non-overlapping index sets:

$$I_0 = \left\{ (i,j) \in \{0, \ldots, n-1\}^2 : \left\lfloor \frac{i}{m} \right\rfloor - \left\lfloor \frac{j}{m} \right\rfloor \in [-1, 1] \right\}, \tag{3}$$

$$I_\ell = \left\{ (i,j) \in \{0, \ldots, n-1\}^2 : \left\lfloor \frac{i}{m_\ell} \right\rfloor - \left\lfloor \frac{j}{m_\ell} \right\rfloor \in \{\pm 2, \pm 3\} \right\}, \quad \ell = 1, \ldots, L. \tag{4}$$

Here the indices start from 0. The index set $I_0$ corresponds to the block tri-diagonals with blocks of size $m$ and the sets $I_\ell$ with $\ell \geq 1$ correspond to block super-diagonals and sub-diagonals. Figure 1b illustrates the structure of $I_0$ (red), $I_1$ (green), and $I_2$ (purple) for $L = 2$ and $m = 2$.

**Fast Multipole Attention** With the hierarchical matrix structure and downsampled keys and values, FMA computes the attention scores at different levels. For a pair $(i,j) \in I_0$, the attention score is calculated with the original $Q$ and $K$. For a pair $(i,j) \in I_\ell$ with $1 \leq \ell \leq L$, the attention score is calculated on level $\ell$ using $Q$ and $K^{(\ell)}$. Specifically, the attention scores are computed via

$$\widetilde{C}_{i,j} = \begin{cases} Q_i K_j^\top & (i,j) \in I_0, \\ Q_i \left( K^{(1)}_{\lfloor jp/m_1 \rfloor} \right)^\top & (i,j) \in I_1, \\ \ldots \\ Q_i \left( K^{(L)}_{\lfloor jp/m_L \rfloor} \right)^\top & (i,j) \in I_L. \end{cases} \tag{5}$$

Here, $Q_i, K_j^{(\ell)}$ denote the $i$th row in $Q$ and the $j$th row in $K^{(\ell)}$, respectively. According to (5), a coarse-level block of size $m_\ell \times m_\ell$ consists of $p$ unique columns, each repeated for $m_\ell/p$ times. Therefore, it can be seen as a rank-$p$ approximation of the corresponding block in the dense attention matrix. $\widetilde{C}$ is a matrix with hierarchical structure as shown in Figure 1b. In causal attention, the upper triangular part of $\widetilde{C}$ is set to $-\infty$. In practice, the full $n \times n$ matrix $\widetilde{C}$ is not formed. Instead, each unique column in the coarse-level blocks is only calculated and stored once.

To normalize attention scores, `softmax` is applied to each row of $\widetilde{C}$,

$$\widetilde{A}_{i,:} = \texttt{softmax}\left( \widetilde{C}_{i,:}/\sqrt{d} \right), \quad i = 0, \ldots, n-1. \tag{6}$$

Next, the attention output on each level is computed by multiplying the normalized attention scores with the values at corresponding locations. Finally, each row of the bidirectional FMA is the sum of outputs from each level:

$$\widetilde{X}_i = \underbrace{\sum_{j:(i,j)\in I_0} \widetilde{A}_{i,j} V_j}_{\text{fine level}} + \underbrace{\sum_{\ell=1}^{L} \sum_{j:(i,j)\in I_\ell} \widetilde{A}_{i,j} V^{(\ell)}_{\lfloor jp/m_\ell \rfloor}}_{\text{coarse levels}}, \quad i = 0, \ldots, n-1. \tag{7}$$

Note that the inner sum in (16) with $m_\ell$ terms contains only $p$ different terms (that are each repeated $m_\ell/p$ times), and hence this sum is implemented efficiently as a sum of $p$ terms.

### 3.3 IMPLEMENTATION AND COMPLEXITY

Implementing (3)-(16) efficiently requires efficient block sparse matrix multiplication and storage, which are not available in standard machine learning frameworks. For faster performance and portability, we build custom CUDA kernels for both forward and backward propagation of FMA with the TVM (Chen et al., 2018) compiler. Our FMA implementation and the cost of each step are as follows:

1. Calculate $K^{(\ell)}$ and $V^{(\ell)}$ on each level by 1D convolution. This requires $\mathcal{O}(npd \log(n/m))$ time and $\mathcal{O}(npd/m)$ space.

2. Calculate dot-products between queries and keys on level $\ell$ at locations in $I_\ell$ as in (5). Then apply softmax across all levels. In each coarse-level block, each unique column is only computed and stored once. This step requires $\mathcal{O}(mnd + npd \log(n/m))$ time and $\mathcal{O}(mn + np \log(n/m))$ space.

3. Calculate attention output on each level and sum up the attention results on each level as in (16). This requires $\mathcal{O}(mnd + npd \log(n/m))$ time and $\mathcal{O}(nd)$ space.

The overall time and space complexity are $\mathcal{O}(mnd + npd \log(n/m))$ and $\mathcal{O}(mn + np \log(n/m))$, where step 2 is dominant.

### 3.4 ANALYSIS OF THE APPROXIMATION ERROR

This section discusses the approximation error of the attention matrix of FMA. For $i \in \{0, \ldots, n-1\}$, let $\widetilde{K}^{(i)} \in \mathbb{R}^{n \times d}$ denote the matrix containing the key vectors at different resolutions corresponding to the $i^{th}$ query (see Equation (5)). For example, the matrix containing the key vectors at different resolutions corresponding to the first $m$ queries $Q_i$, $i = 0, \ldots, m_1$ and $p = 1$ is given by

$$\widetilde{K}^{(i)} = \left[ \underbrace{K_0^\top, \ldots, K_{2m-1}^\top}_{\text{fine level}}, \underbrace{K^{(1)\top}_{\left\lfloor \frac{2m}{m_1} \right\rfloor}, \ldots, K^{(1)\top}_{\left\lfloor \frac{3m}{m_1} \right\rfloor}}_{m_1 \text{ columns}}, \ldots, \underbrace{K^{(1)\top}_{\left\lfloor \frac{3m}{m_1} \right\rfloor}, \ldots}_{m_1 \text{ columns}}, \underbrace{K^{(2)\top}_{\left\lfloor \frac{4m}{m_2} \right\rfloor}, \ldots}_{m_2 \text{ columns}}, \underbrace{K^{(2)\top}_{\left\lfloor \frac{6m}{m_2} \right\rfloor}, \ldots}_{m_2 \text{ columns}}, \ldots, \ldots \right]^\top \in \mathbb{R}^{n \times d} \quad (8)$$

where each column on coarse levels is repeated for $m_\ell$ times. Since $\widetilde{C}_i = Q_i \widetilde{K}^{(i),T} \in \mathbb{R}^{1 \times n}$, we have

$$\widetilde{C} = \boldsymbol{e}_1 \widetilde{C}_0 + \cdots + \boldsymbol{e}_n \widetilde{C}_{n-1} = \boldsymbol{e}_1 Q_0 \widetilde{K}^{(0),T} + \cdots + \boldsymbol{e}_n Q_{n-1}^T \widetilde{K}^{(n-1),T}, \quad (9)$$

where $\{\boldsymbol{e}_1, \ldots, \boldsymbol{e}_n\}$ is the standard basis for $\mathbb{R}^n$. Then we have the following error estimations of the approximation matrix and its exponential. The detailed derivation is given in the Appendix A.4.

**Theorem 1** (Error Estimation of the Attention Matrix Approximation). *Let $\widetilde{C}$ be our two-level FMA attention matrix with $p = 1$. Suppose for every $j, \ell \in \{0, \ldots, m-1\}, t \in \{0, 1, 2, 3\}$, and $i \in \{0, \ldots, n-1\}$, we have*

$$\|Q_i^T\|_2 \leq \beta_1 \quad and \quad \|K_{tm+j}^T - K_{tm+\ell}^T\|_2 \leq \beta_2, \quad (10)$$

*and the kernel weights are normalized in the sense that $\sum_{j=0}^{m-1} \alpha_j = 1$. Then*

$$\|\widetilde{C} - C\|_F \leq m \beta_1 \beta_2 \sqrt{6 \sum_{j=0}^{m-1} \alpha_j^2}. \quad (11)$$

Note that the results can be easily extended to higher level $L > 1$ and $p > 1$. In addition, inspired by the estimation derived in Zeng et al. (2022), we have a similar estimation of $\|\exp(\widetilde{C}) - \exp(C)\|_F$ where our set of components for the approximation is obtained from the hierarchical structure (see Fig. 1b).

**Theorem 2.** *Let $\widetilde{C}$ be the approximated attention matrix obtained from our $(L+1)$-level FMA with fixed kernel weights and $\delta_k$ be the $m_k$-th largest coefficient. Suppose*

$$\|Q_i^T\|_2 \leq \beta_1 \quad and \quad \|K_{j_1}^T - K_{j_2}^T\|_2 \leq \beta_2, \quad (12)$$

*for $(i, j_1)$ and $(i, j_2)$ belong to the same level $I_\ell$, for $\ell = 1, \ldots L$ and $i \in \{0, \ldots, n-1\}$. Then*

$$\| \exp(\widetilde{C}) - \exp(C) \|_F^2 \leq \sum_{\ell=1}^{L} C_{2r} \delta_k^2 \left(3nm_k - 6m_k^2\right).$$
(13)

*Here $r = 2\beta_1\beta_2$ and $C_{2r} = 1 + \exp(2r) - \exp(r)$.*

It is worth noting that if $\delta_k$ decays fast as $k$ increases (which we can see in fast multipole method), the error in Eq.13 becomes small.

## 4 EXPERIMENTS

In this section, we experimentally analyze the performance of the proposed method. First, we evaluate FMA through autoregressive language modeling on the `enwik-8` dataset ((Mahoney, 2009), Sec. 4.1). Then we evaluate FMA in the bidirectional setting through masked language modeling on `Wikitext-103` ((Merity et al., 2017), Sec. 4.2). In both tasks, we compare FMA with efficient attention variants used in other works focusing on improving the efficiency of attention for long sequences, including Memory-compressed attention (Liu et al., 2018), Reformer (Kitaev et al., 2020), linear attention (Katharopoulos et al., 2020), Nyströmformer (Xiong et al., 2021), BigBird (Zaheer et al., 2020), H-transformer (Zhu & Soricut, 2021), and MRA (Zeng et al., 2022). The context size varies from 512 to 4K. For fair comparison, we use the same transformer architecture for all attention variants. Finally, in Sec. 4.3, we show a comparative analysis of memory and time cost to demonstrate the ability of FMA to scale efficiently.

We use $p = 4$ in our experiments for Fast Multipole Attention and typically use 3, 4 or 5 levels (including the finest level). In general, increasing $m$ and $p$ leads to better accuracy at the cost of larger computational resources (see more detail in the supplementary material). For Reformer, we use two rounds of hashing and the number of buckets is set to the nearest power of two that is greater than or equal to the square root of the input size. We set the numerical rank to be 16 in H-transformer. We conduct the experiments on one NVIDIA Tesla A100 with 80GB of memory and use the `fairseq` toolkit (Ott et al., 2019).

### 4.1 AUTOREGRESSIVE LANGUAGE MODELING

We consider the `enwik-8` dataset, which contains the first $10^8$ bytes of the English Wikipedia dump from 2006. We train a character-level autoregressive language model that estimates the distribution of the next token given previous tokens. We adopt the standard transformer decoder with pre-normalization and learned positional embeddings, and replace full attention with other attention variants to compare their performance. The model has 6 layers with embedding dimension 768 and 12 attention heads.

Table 1 shows that with the same context size, FMA obtains lower bpc using similar or less memory than other variants of efficient attention. Not surprisingly, full attention is more accurate than the efficient variants but requires quadratic memory which is unfeasible for long sequences. Also, FMA-linear uses less memory than FMA while achieving comparable bpc. Models with quadratic complexity are highlighted in maroon. Compared to H-transformer, FMA obtains dramatically better accuracy with smaller memory size.

### 4.2 BIDIRECTIONAL LANGUAGE MODEL

We conduct bidirectional language modeling on `Wikitext-103` (Merity et al., 2017), a dataset commonly used for testing long term dependencies in language models, see Table 2. We process `Wikitext-103` with Byte Part Encoding tokenization and train a bidirectional language model by randomly masking 15% of the input tokens. We use the Transformer encoder with 6 layers, embedding dimension 768 and 12 attention heads, and we compare the performance and memory cost of different efficient attention variants, including our new FMA, by replacing the full attention module in the Transformer. The results in Table 2 show that our new FMA obtains the best accuracy next to the full attention with low memory footprint compared to other efficient transformers from the literature.

| model | context size | bpc(test)↓ | memory footprint |
|---|---|---|---|
| Full | 512 | 1.346 | 16.5 GB |
| Compressed (Liu et al., 2018) | 512 | 1.439 | 13.5 GB |
| Reformer (Kitaev et al., 2020) | 512 | 1.372 | 15.5 GB |
| Linear (Katharopoulos et al., 2020) | 512 | 1.424 | 38.2 GB |
| Nyströmformer (Xiong et al., 2021) | 512 | 1.418 | 14.2 GB |
| BigBird (Zaheer et al., 2020) | 512 | 1.465 | 16.4 GB |
| H-transformer (Zhu & Soricut, 2021) | 512 | 1.853 | 12.3 GB |
| FMA-m=64-3lev | 512 | **1.353** | 11.6 GB |
| FMA-linear-m=64-3lev | 512 | 1.391 | **9.8 GB** |
| Full | 1024 | 1.236 | 46.5 GB |
| Compressed (Liu et al., 2018) | 1024 | 1.351 | 27.4 GB |
| Reformer (Kitaev et al., 2020) | 1024 | 1.271 | 31.8 GB |
| Linear (Katharopoulos et al., 2020) | 1024 | 1.343 | 74.9 GB |
| Nyströmformer (Xiong et al., 2021) | 1024 | 1.290 | 26.9 GB |
| BigBird (Zaheer et al., 2020) | 1024 | 1.345 | 23.1 GB |
| H-transformer (Zhu & Soricut, 2021) | 1024 | 1.789 | 23.7 GB |
| FMA-m=64-4lev | 1024 | **1.256** | 21.6 GB |
| FMA-linear-m=64-4lev | 1024 | 1.298 | **17.6 GB** |
| Full | 2048 | 1.128 | 2×71.5 GB |
| Compressed (Liu et al., 2018) | 2048 | 1.276 | 73.4 GB |
| Nyströmformer (Xiong et al., 2021) | 2048 | 1.224 | 67.1 GB |
| H-transformer (Zhu & Soricut, 2021) | 2048 | 1.684 | 38.4 GB |
| FMA-m=128-4lev | 2048 | **1.220** | 48.2 GB |
| FMA-linear-m=128-4lev | 2048 | 1.244 | **34.1 GB** |
| Full | 4096 | 1.091 | 8×61.7 GB |
| H-transformer (Zhu & Soricut, 2021) | 4096 | 1.535 | 66.0 GB |
| FMA-linear-m=128-5lev | 4096 | 1.154 | 69.4 GB |

Table 1: Autoregressive language modeling results on enwik-8. With the same context size, FMA obtains lower bpc using less memory than other variants of efficient attention. For full attention with sequence lengths 2048 and 4096 we have to reduce the batch size and accumulate the gradient over batches to avoid GPU memory overflow, as listed in the accumulated memory use. Bold numbers are best among the efficient transformers (not counting the $\mathcal{O}(n^2)$ transformers).

## 4.3 EFFICIENCY COMPARISON

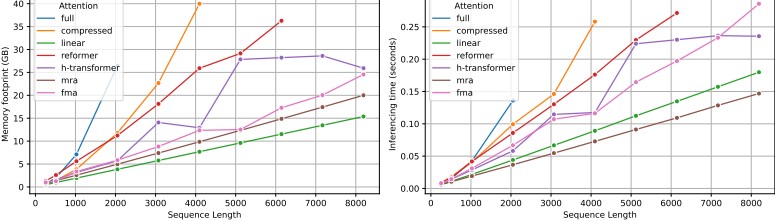

Figure 2: Efficiency comparison of attention variants on different input sizes.

In order to demonstrate the scalability of FMA on long sequences, we compare its inference time and GPU memory footprint to those of full attention, memory compressed attention, MRA attention, LSH attention, linear attention, and H-attention. The input size varies from 256 to 8K and the batch size, embedding dimension, and number of attention heads are 64, 768, and 12, respectively.

As shown in Figure 2, the memory cost of FMA scales much better than full attention and is lower than or similar to that of other efficient variants. Similarly, the inference speed scales much better

| model | context size | ppl(test)↓ | memory footprint |
|---|---|---|---|
| Full | 512 | 9.21 | 15.4 GB |
| Reformer (Kitaev et al., 2020) | 512 | 11.03 | 14.6 GB |
| Linear (Katharopoulos et al., 2020) | 512 | 14.95 | 11.8 GB |
| MRA (Zeng et al., 2022) | 512 | 11.82 | 11.1 GB |
| H-transformer (Zhu & Soricut, 2021) | 512 | 10.60 | 10.7 GB |
| FMA-m=64-3lev | 512 | **9.45** | 10.9 GB |
| FMA-linear-m=64-3lev | 512 | 10.78 | **8.2 GB** |
| Full | 1024 | 8.89 | 36.8 GB |
| Reformer (Kitaev et al., 2020) | 1024 | 10.56 | 27.4 GB |
| Linear (Katharopoulos et al., 2020) | 1024 | 14.24 | 22.0 GB |
| MRA (Zeng et al., 2022) | 1024 | 10.20 | 17.5 GB |
| H-transformer (Zhu & Soricut, 2021) | 1024 | 9.81 | 16.9 GB |
| FMA-m=64-4lev | 1024 | **9.06** | 18.6 GB |
| FMA-linear-m=64-4lev | 1024 | 10.13 | **15.6 GB** |
| Full | 2048 | 8.70 | $2\times56.5$ GB |
| Reformer (Kitaev et al., 2020) | 2048 | 9.90 | 49.2 GB |
| Linear (Katharopoulos et al., 2020) | 2048 | 14.06 | 48.1 GB |
| MRA (Zeng et al., 2022) | 2048 | 9.66 | 35.3 GB |
| H-transformer (Zhu & Soricut, 2021) | 2048 | 9.45 | 33.1 GB |
| FMA-m=64-5lev | 2048 | **8.95** | 37.0 GB |
| FMA-linear-m=64-5lev | 2048 | 9.64 | **29.5 GB** |

Table 2: Masked language modeling results on `Wikitext-103`. With the same context size, our FMA obtains the best accuracy and uses less memory compared to other efficient attention variants.

than full attention and is similar to that of other efficient variants. However, we should note that our current CUDA kernels are not yet fully optimized and that more efficient implementation of the kernels could further improve the time and memory efficiency.

## 5 CONCLUSION

We have presented Fast Multipole Attention, an efficient variant of dot-product attention. With hierarchical grouping and downsampling of queries and keys, FMA achieves $\mathcal{O}(n \log n)$ or $\mathcal{O}(n)$ complexity while retaining a global receptive field. FMA is similar in spirit to recently proposed hierarchical attention mechanisms such as H-transformer (Zhu & Soricut, 2021) and Multi-Resolution Analysis (MRA) attention (Zeng et al., 2022) but the key advance is that we *learn* the basis functions that compute group quantities, as opposed to the fixed averaging basis functions of H-transformer and the fixed wavelet family used in MRA attention. In both autoregressive and bidirectional tasks, our FMA outperforms other efficient attention variants overall. In particular, our numerical results show that *learning the basis functions* leads to substantially better accuracy for language tasks compared to H-transformer and MRA attention, for the same memory and time cost. FMA is versatile and robust as it can be used as a drop-in replacement of causal or bidirectional attention without changing the model architecture or the training schedule. The high efficiency of FMA has the potential to empower large language models with much greater sequence lengths, taking the full context into account in a naturally hierarchical manner during training and when generating long sequences. Our findings strongly suggest that future work combining our approach in FMA to learn the basis functions with the adaptivity of multi-resolution methods like the MRA attention from (Zeng et al., 2022) may lead to the next quantum leap in designing accurate, efficient and versatile multilevel transformer methods in the future. As for the Fast Multipole Method, the algorithmic principle of Fast Multipole Attention can naturally be extended to arrays in multiple dimensions including images and videos, so our new Fast Multipole Attention approach promises to have wide applications beyond language modeling.

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

## A    SUPPLEMENTARY MATERIAL

### A.1    NOTATION

For a vector $\boldsymbol{z} = \begin{bmatrix} z_0 & z_1 & \ldots & z_{n-1} \end{bmatrix}^T \in \mathbb{R}^n$, the $\ell_1$-norm and the $\ell_\infty$-norm of $\boldsymbol{z}$ are given by

$$\|\boldsymbol{z}\|_1 = \sum_{j=0}^{n-1} |z_j| \quad \text{and} \quad \|\boldsymbol{z}\|_\infty = \max_{0 \le j \le n-1} |z_j|.$$

For a matrix $W \in \mathbb{R}^{m \times n}$, $\|W\|_\infty$ denotes the maximum absolute row sum of the matrix $W$,

$$\|W\|_\infty = \max_{0 \le i \le m-1} \sum_{j=0}^{n-1} |W_{ij}| = \sup_{\boldsymbol{z} \ne \vec{0}} \frac{\|W\boldsymbol{z}\|_\infty}{\|\boldsymbol{z}\|_\infty},$$

and $\|W\|_1$ denotes the maximum absolute column sum of the matrix $W$,

$$\|W\|_1 = \max_{0 \le j \le n-1} \sum_{i=0}^{m-1} |W_{ij}| = \sup_{\boldsymbol{z} \ne \vec{0}} \frac{\|W\boldsymbol{z}\|_1}{\|\boldsymbol{z}\|_1}.$$

### A.2    FMA-LINEAR

FMA-linear is described as follow. First, queries, keys, and values are downsampled to get $Q^{(\ell)}$, $K^{(\ell)}$, and $V^{(\ell)}$ for $\ell = 1, \ldots, L$. The attention scores are computed by

$$\widetilde{C}_{i,j} = \begin{cases} Q_i K_j^\top & (i,j) \in I_0, \\ \left( Q^{(1)}_{\lfloor ip/m_1 \rfloor} \right) \left( K^{(1)}_{\lfloor jp/m_1 \rfloor} \right)^\top & (i,j) \in I_1, \\ \ldots \\ \left( Q^{(L)}_{\lfloor ip/m_L \rfloor} \right) \left( K^{(L)}_{\lfloor jp/m_L \rfloor} \right)^\top & (i,j) \in I_L. \end{cases} \tag{14}$$

For $\ell = 0, \ldots, L$ and each entry $(i,j) \in I_\ell$, softmax is computed at each level:

$$\widetilde{A}_{i,j} = \frac{\exp\left( \widetilde{C}_{i,j}/\sqrt{d} \right)}{\sum_{(i,k) \in I_\ell} \exp\left( \widetilde{C}_{i,k}/\sqrt{d} \right)}. \tag{15}$$

Finally, the output is given by

$$\widetilde{X}_i = \sum_{j:(i,j) \in I_0} \widetilde{A}_{i,j} V_j + \sum_{\ell=1}^{L} \sum_{j:(i,j) \in I_\ell} \widetilde{A}_{i,j} V^{(\ell)}_{\lfloor jp/m_\ell \rfloor}, \quad i = 0, \ldots, n-1. \tag{16}$$

### A.3    OVERVIEW OF BIDIRECTIONAL FMA

Figure 3 gives an overview of FMA with three levels.

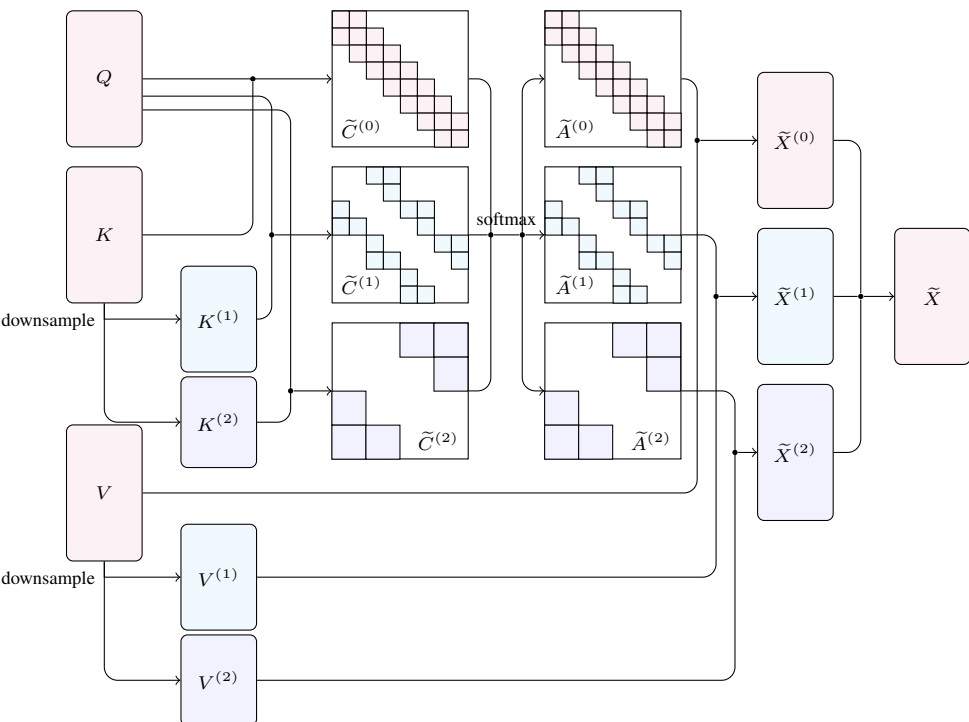

Figure 3: Overview of bidirectional FMA with three levels.

## A.4 THEORETICAL DERIVATION

This section gives the detailed proof of our theoretical results. Specifically, we have the following two theorems to estimate the errors of the attention matrix approximation $\|\widetilde{C} - C\|_F$ and $\|\exp(\widetilde{C}) - \exp(C)\|_F$. The latter estimation is inspired by the estimation derived in Zeng et al. (2022) where our set of components for the approximation is obtained from the hierarchical structure (see Fig. 1b).

**Theorem 3** (Error Estimation of the Attention Matrix Approximation). *Let $\widetilde{C}$ be our two-level FMA attention matrix with $p = 1$. Suppose for every $j, \ell \in \{0, \dots, m-1\}, t \in \{0, 1, 2, 3\}$, and $i \in \{0, \dots, n-1\}$, we have*

$$\|Q_i^T\|_2 \leq \beta_1 \quad and \quad \|K_{tm+j}^T - K_{tm+\ell}^T\|_2 \leq \beta_2, \tag{17}$$

*and the kernel weights are normalized in the sense that $\sum_{j=0}^{m-1} \alpha_j = 1$. Then*

$$\|\widetilde{C} - C\|_F \leq m\beta_1\beta_2 \sqrt{6 \sum_{j=0}^{m-1} \alpha_j^2}. \tag{18}$$

*Proof.* In a two-level FMA, we have $L = 1$ and $n = 4m$. For $i = 0, \ldots, m - 1$, we have

$$
\widetilde{K}^{(i)} = \begin{bmatrix} K_0 \\ \vdots \\ K_{2m-1} \\ \sum\limits_{j=0}^{m-1} \alpha_j K_{2m+j} \\ \vdots \\ \sum\limits_{j=0}^{m-1} \alpha_j K_{2m+j} \\ \sum\limits_{j=0}^{m-1} \alpha_j K_{3m+j} \\ \vdots \\ \sum\limits_{j=0}^{m-1} \alpha_j K_{3m+j} \end{bmatrix}, \quad \widetilde{K}^{(3m+i)} = \begin{bmatrix} \sum\limits_{j=0}^{m-1} \alpha_j K_j \\ \vdots \\ \sum\limits_{j=0}^{m-1} \alpha_j K_j \\ \sum\limits_{j=0}^{m-1} \alpha_j K_{m+j} \\ \vdots \\ \sum\limits_{j=0}^{m-1} \alpha_j K_{m+j} \\ K_{2m} \\ \vdots \\ K_{4m-1} \end{bmatrix},
\tag{19}
$$

$$
\widetilde{K}^{(m+i)} = \begin{bmatrix} K_0 \\ \vdots \\ K_{3m-1} \\ \sum\limits_{j=0}^{m-1} \alpha_j K_{3m+j} \\ \vdots \\ \sum\limits_{j=0}^{m-1} \alpha_j K_{3m+j} \end{bmatrix}, \quad \widetilde{K}^{(2m+i)} = \begin{bmatrix} \sum\limits_{j=0}^{m-1} \alpha_j K_j \\ \vdots \\ \sum\limits_{j=0}^{m-1} \alpha_j K_j \\ K_m \\ \vdots \\ K_{4m-1} \end{bmatrix}.
$$

Then for $i = 0, \ldots, m - 1$, using Cauchy-Schwarz inequality, we have

$$\|\widetilde{C}_i - C_i\|_F^2 = \|Q_i(\widetilde{K}^{(i)} - K)^T\|_F^2 = \|(\widetilde{K}^{(i)} - K)Q_i^T\|_2^2$$

$$= \left\| \begin{bmatrix} 0_{2m \times 1} \\ \sum_{j=0}^{m-1} \alpha_j(K_{2m+j} - K_{2m})Q_i^T \\ \vdots \\ \sum_{j=0}^{m-1} \alpha_j(K_{2m+j} - K_{3m-1})Q_i^T \\ \sum_{j=0}^{m-1} \alpha_j(K_{3m+j} - K_{3m})Q_i^T \\ \vdots \\ \sum_{j=0}^{m-1} \alpha_j(K_{3m+j} - K_{4m-1})Q_i^T \end{bmatrix} \right\|_2^2$$

$$= \left( \sum_{j=0}^{m-1} \alpha_j(K_{2m+j} - K_{2m})Q_i^T \right)^2 + \cdots + \left( \sum_{j=0}^{m-1} \alpha_j(K_{2m+j} - K_{3m-1})Q_i^T \right)^2 +$$

$$\left( \sum_{j=0}^{m-1} \alpha_j(K_{3m+j} - K_{3m})Q_i^T \right)^2 + \cdots + \left( \sum_{j=0}^{m-1} \alpha_j(K_{3m+j} - K_{4m-1})Q_i^T \right)^2$$

$$\leq \|Q_i^T\|_2^2 \left( \left\| \sum_{j=0}^{m-1} \alpha_j(K_{2m+j} - K_{2m})^T \right\|_2^2 + \cdots + \left\| \sum_{j=0}^{m-1} \alpha_j(K_{2m+j} - K_{3m-1})^T \right\|_2^2 + \right.$$

$$\left. \left\| \sum_{j=0}^{m-1} \alpha_j(K_{3m+j} - K_{3m})^T \right\|_2^2 + \cdots + \left\| \sum_{j=0}^{m-1} \alpha_j(K_{3m+j} - K_{4m-1})^T \right\|_2^2 \right)$$

$$\leq \|Q_i^T\|_2^2 \left( \sum_{j=0}^{m-1} \alpha_j^2 \right) \left( \sum_{j=0}^{m-1} \left\| (K_{2m+j} - K_{2m})^T \right\|_2^2 + \cdots + \sum_{j=0}^{m-1} \left\| (K_{2m+j} - K_{3m-1})^T \right\|_2^2 + \right.$$

$$\left. \sum_{j=0}^{m-1} \left\| (K_{3m+j} - K_{3m})^T \right\|_2^2 + \cdots + \sum_{j=0}^{m-1} \left\| (K_{3m+j} - K_{4m-1})^T \right\|_2^2 \right)$$

$$\leq (2m^2\beta_2^2)\|Q_i^T\|_2^2 \left( \sum_{j=0}^{m-1} \alpha_j^2 \right).$$

In conclusion, we have

$$\|\widetilde{C}_i - C_i\|_F^2 \leq (2m^2\beta_2^2)\|Q_i^T\|_2^2 \left( \sum_{j=0}^{m-1} \alpha_j^2 \right) \leq 2m^2\beta_1^2\beta_2^2 \left( \sum_{j=0}^{m-1} \alpha_j^2 \right), \quad \text{for} \quad i = 0, \ldots, m - 1. \tag{20}$$

Similarly,

$$\|\widetilde{C}_{3m+i} - C_{3m+i}\|_F^2 \leq (2m^2\beta_2^2)\|Q_{3m+i}^T\|_2^2 \left( \sum_{j=0}^{m-1} \alpha_j^2 \right) \leq 2m^2\beta_1^2\beta_2^2 \left( \sum_{j=0}^{m-1} \alpha_j^2 \right) \tag{21}$$

$$\|\widetilde{C}_{m+i} - C_{m+i}\|_F^2 \leq (m^2\beta_2^2)\|Q_{m+i}^T\|_2^2 \left( \sum_{j=0}^{m-1} \alpha_j^2 \right) \leq m^2\beta_1^2\beta_2^2 \left( \sum_{j=0}^{m-1} \alpha_j^2 \right) \tag{22}$$

$$\|\widetilde{C}_{2m+i} - C_{2m+i}\|_F^2 \leq (m^2\beta_2^2)\|Q_{2m+i}^T\|_2^2 \left( \sum_{j=0}^{m-1} \alpha_j^2 \right) \leq m^2\beta_1^2\beta_2^2 \left( \sum_{j=0}^{m-1} \alpha_j^2 \right). \tag{23}$$

So

$$\|\widetilde{C} - C\|_F^2 = \sum_{i=0}^{4m-1} \|\widetilde{C}_i - C_i\|_F^2 \leq (6m^2\beta_1^2\beta_2^2)\left(\sum_{j=0}^{m-1} \alpha_j^2\right).$$

□

**Theorem 4.** *Let $\widetilde{C}$ be the approximated attention matrix obtained from our $(L+1)$-level FMA with fixed kernel weights and $\delta_k$ be the $m_k$-th largest coefficient. Suppose*

$$\|Q_i^T\|_2 \leq \beta_1 \quad and \quad \|K_{j_1}^T - K_{j_2}^T\|_2 \leq \beta_2, \tag{24}$$

*for $(i, j_1)$ and $(i, j_2)$ belong to the same level $I_\ell$, for $\ell = 1, \ldots L$ and $i \in \{0, \ldots, n-1\}$. Then*

$$\|\exp(\widetilde{C}) - \exp(C)\|_F^2 \leq \sum_{\ell=1}^{L} C_{2r}\delta_k^2\left(3nm_k - 6m_k^2\right). \tag{25}$$

*Here $r = 2\beta_1\beta_2$ and $C_{2r} = 1 + \exp(2r) - \exp(r)$.*

*Proof.* Following the arguments in Proposition A.3. in Zeng et al. (2022), we would like to bound $\|\exp(\widetilde{C}) - \exp(C)\|_F^2$ for our multilevel FMA. The number of blocks of each level is given below:

- At the fine level (the red blocks in Fig 1b, we have $\left(3\dfrac{n}{m_1} - 2\right)$ blocks of size $m_1 \times m_1$.

- At the first averaging level (the green blocks in Fig 1b, we have $\left(3\dfrac{n}{m_1} - 6\right)$ blocks of size $m_1 \times m_1$.

- At the second averaging level (the purple blocks in Fig 1b, we have $\left(3\dfrac{n}{m_2} - 6\right)$ blocks of size $m_2 \times m_2$.

- At the $L^{th}$ averaging level, we have $\left(3\dfrac{n}{m_L} - 6\right) = 6$ blocks of size $m_L \times m_L$.

Similar to the arguments in Proposition A.3. in Zeng et al. (2022), counting the sets of potential components in the approximation yields

$$\|\exp(\widetilde{C}) - \exp(C)\|_F^2 \leq C_{2r}\delta_1^2 \sum_{1-st\,level} m_1^2 + \cdots + C_{2r}\delta_L^2 \sum_{L-th\,level} m_L^2$$

$$\leq \sum_{\ell=1}^{L} C_k\delta_k^2\left(3\frac{n}{m_k} - 6\right)m_k^2$$

$$\leq \sum_{\ell=1}^{L} C_{2r}\delta_k^2\left(3nm_k - 6m_k^2\right).$$

□

## A.5 DIFFERENT SETTINGS OF THE FINE LEVEL SIZE $m$ AND AND THE RANK $p$

To explore the effect of changing the fine level size $m$ and and the rank $p$ in FMA, we conduct experiments on enwik-8 and measure the bpc and the memory cost. Table 3 shows that increasing $m$ and $p$ gives lower bpc at the cost of an increased memory footprint. Similar to Section 4.1, the model has 6 layers with embedding dimension 768 and 12 attention heads. The memory cost for storing the attention matrix with hierarchical structure is $\mathcal{O}(mn + np\log(n/m))$, which is the dominant part of the total memory footprint for large sequence lengths. In the main paper we consistently use $p = 4$, which gives a good balance between accuracy and memory use.

|          | m=16 (6 levels)   | m=32 (5 levels)   | m=64 (4 levels)   |
|----------|-------------------|-------------------|-------------------|
| p=2      | 1.320 (18.4 GB)   | 1.272 (19.3 GB)   | 1.258 (20.9 GB)   |
| p=4      | 1.307 (19.1 GB)   | 1.266 (19.7 GB)   | 1.256 (21.6 GB)   |
| p=8      | 1.276 (20.2 GB)   | 1.262 (21.1 GB)   | 1.251 (23.2 GB)   |

Table 3: Autoregressive language modeling results with different $m$ and $p$ on enwik-8 with context size 1024.

### A.6  TVM TENSOR EXPRESSIONS

We implement FMA using TVM (Chen et al., 2018) in four stages. This section contains the mathematical expressions used by the TVM compilers:

1. Local outgoing expansion, which calculates the attention locally on the fine level.
2. Far-field outgoing expansion, which calculates the attention at far away locations on coarse levels.
3. Local incoming expansion, which gathers the values from the neighborhood on the fine level.
4. Far-field incoming expansion, which gathers the values from far away locations on coarse levels.

For simplicity, we denote $\frac{\partial \mathcal{L}}{\partial M_{ij}}$ by $\partial M(i,j)$ for a scalar $\mathcal{L}$ and matrix $M$.

#### A.6.1  LOCAL OUTGOING EXPANSION

Forward: For $Q \in \mathbb{R}^{n \times d}, K \in \mathbb{R}^{n \times d}$, the output matrix $Z \in \mathbb{R}^{n \times 2m}$ is given by

$$Z(i,j) = \sum_{k=0}^{d-1} Q(i,k)K(m\lfloor i/m \rfloor + j, k). \tag{26}$$

Backward: For $\partial Z \in \mathbb{R}^{n \times 2m}, K \in \mathbb{R}^{n \times d}$, the partial gradient $\partial Q \in \mathbb{R}^{n \times d}$ is given by

$$\partial Q(i,j) = \sum_{k=0}^{2m-1} \partial Z(i,k)K(m\lfloor i/m \rfloor + k, j). \tag{27}$$

For $\partial Z \in \mathbb{R}^{n \times 2m}, Q \in \mathbb{R}^{n \times d}$, the partial gradient $\partial K \in \mathbb{R}^{n \times d}$ is given by

$$\partial K(i,j) = \sum_{k=0}^{m-1} \partial Z(m\lfloor i/m \rfloor + k, i\%m + m)Q(m\lfloor i/m \rfloor + k, j)$$
$$+ \sum_{k=m}^{2m-1} \partial Z(m\lfloor i/m \rfloor + k, i\%m)Q(m\lfloor i/m \rfloor + k, j). \tag{28}$$

#### A.6.2  FAR-FIELD OUTGOING EXPANSION

Forward: For $Q \in \mathbb{R}^{n \times d}, K \in \mathbb{R}^{n_\ell \times d}$, the output matrix $Z \in \mathbb{R}^{n \times 2p}$ is given by

$$Z(i,j) = \sum_{k=0}^{d-1} Q(i,k)K(2p\lfloor i/2w \rfloor + j, k), \tag{29}$$

with $w \coloneqq p\frac{n}{n_l}$.

Backward: For $\partial Z \in \mathbb{R}^{n \times 2p}, K \in \mathbb{R}^{n_\ell \times d}$, the partial gradient $\partial Q \in \mathbb{R}^{n \times d}$ is given by

$$\partial Q(i,j) = \sum_{k=0}^{2p-1} \partial Z(i,k)K(2p\lfloor i/2w \rfloor + k, j). \tag{30}$$

For $\partial Z \in \mathbb{R}^{n \times 2p}, Q \in \mathbb{R}^{n \times d}$, the partial gradient $\partial K \in \mathbb{R}^{n_\ell \times d}$ is given by

$$\partial K(i,j) = \sum_{k=0}^{2w-1} \partial Z(2w(\lfloor i/2p \rfloor + 1) + k, i\%2p)Q(2w(\lfloor i/2p \rfloor + 1) + k, j). \tag{31}$$

### A.6.3 LOCAL INCOMING EXPANSION

Forward: For $C \in \mathbb{R}^{n \times 2m}, V \in \mathbb{R}^{n \times d}$, the output matrix $Z \in \mathbb{R}^{n \times d}$ is given by

$$Z(i,j) = \sum_{k=0}^{2m-1} C(i,k)V(m\lfloor i/m \rfloor + k, j). \tag{32}$$

Backward: For $\partial Z \in \mathbb{R}^{n \times d}, V \in \mathbb{R}^{n \times d}$, the partial gradient $\partial C \in \mathbb{R}^{n \times 2m}$ is given by

$$\partial C(i,j) = \sum_{k=0}^{d-1} \partial Z(i,k)V(m\lfloor i/m \rfloor + j, k). \tag{33}$$

For $\partial Z \in \mathbb{R}^{n \times d}, C \in \mathbb{R}^{n \times 2m}$, the partial gradient $\partial V \in \mathbb{R}^{n \times d}$ is given by

$$\partial V(i,j) = \sum_{k=0}^{m-1} \partial Z(m\lfloor i/m \rfloor + k, j)C(m\lfloor i/m \rfloor + k, i\%m + m)$$
$$+ \sum_{k=m}^{2m-1} \partial Z(m\lfloor i/m \rfloor + k, j)C(m\lfloor i/m \rfloor + k, i\%m). \tag{34}$$

### A.6.4 FAR-FIELD INCOMING EXPANSION

Forward: For $C \in \mathbb{R}^{n \times 2p}, V \in \mathbb{R}^{n_\ell \times d}$, the output matrix $Z \in \mathbb{R}^{n \times d}$ is given by

$$Z(i,j) = \sum_{k=0}^{2p-1} C(i,k)V(2p\lfloor i/2w \rfloor + k, j). \tag{35}$$

Backward: For $\partial Z \in \mathbb{R}^{n \times d}, V \in \mathbb{R}^{n_\ell \times d}$, the partial gradient $\partial C \in \mathbb{R}^{n \times 2p}$ is given by

$$\partial C(i,j) = \sum_{k=0}^{d-1} \partial Z(i,k)V(2p\lfloor i/2w \rfloor + j, k). \tag{36}$$

For $\partial Z \in \mathbb{R}^{n \times d}, C \in \mathbb{R}^{n \times 2p}$, the partial gradient $\partial V \in \mathbb{R}^{n_\ell \times d}$ is given by

$$\partial V(i,j) = \sum_{k=0}^{2w-1} \partial Z(2w(\lfloor i/2p \rfloor + 1) + k, j)C(2w(\lfloor i/2p \rfloor + 1) + k, i\%2p). \tag{37}$$

## A.7 RESULTS OF LARGE MODELS WITH 12 ATTENTION LAYERS FOR AUTOREGRESSIVE LANGUAGE MODELING

To validate FMA in larger models, we conduct the autoregressive language modeling experiments in section 4.1 with transformer model of 12 layers. The results are shown in table 4.s

## A.8 RESULTS ON LONG RANGE ARENA BENCHMARK

Long range arena (Tay et al., 2021) is a benchmark specifically designed for efficient transformers with long input sequences. The benchmark consists of five tasks: Long sequence ListOps, Byte-level text classification, document retrieval, image classification, and Pathfinder. The sequence length varies from 256 to 8K. Table 5 shows that FMA and FMA-linear achieve competitive results across all the tasks among the efficient transformer models. One limitation of long range arena is that it only includes bidirectional tasks. Also, they are all classification problems. Listops, retrieval and text are for 1D data and as expected, FMA performs well. Pathfinder and image are 2D, where FMA is not expected to perform well. MRA performs better than FMA for 2D data due to the adaptivity of MRA. Due to these limitations, the long range arena benchmark is not fully representative for the language modelling tasks we describe in the main paper, in particular, the autoregressive task.

| model | context size | bpc(test)↓ | memory footprint |
|---|---|---|---|
| Full-large | 1024 | 1.188 | 2×42.7 GB |
| FMA-large-m=64-4lev | 1024 | 1.242 | 44.0 GB |

Table 4: Results of large models with 12 attention layers. For full attention, we have to reduce the batch size from 32 to 16 and accumulate the gradient over batches to avoid GPU memory overflow, as listed in the accumulated memory use.

| model | listops | pathfinder | retrieval | text | image |
|---|---|---|---|---|---|
| Full | 0.3901 | 0.7031 | 0.8045 | 0.6011 | 0.3687 |
| Nyströmformer | 0.3765 | 0.6678 | 0.8049 | 0.6153 | 0.3985 |
| Skyformer | 0.3876 | 0.7011 | 0.8132 | 0.6037 | 0.3293 |
| Reformer | 0.3679 | 0.6212 | 0.7877 | 0.6078 | 0.4389 |
| Performer | 0.3765 | 0.6838 | 0.7955 | 0.6024 | 0.3744 |
| BigBird | 0.3841 | 0.6838 | 0.8058 | 0.6098 | 0.3573 |
| H-transformer | 0.3744 | 0.6768 | 0.7837 | 0.6470 | 0.4203 |
| MRA | 0.3841 | 0.6668 | 0.7781 | 0.6246 | 0.3737 |
| FMA | 0.3881 | 0.6413 | 0.8021 | 0.6152 | 0.3565 |
| FMA-linear | 0.3791 | 0.6243 | 0.8002 | 0.5908 | 0.3540 |

Table 5: Long range arena accuracy results.

## A.9 FAST MULTIPOLE METHODS

In this section, we draw connections between the attention mechanism and $n$-body problems and recall the Fast Multipole Method (Greengard & Rokhlin, 1987; Beatson & Greengard, 1997; Martinsson, 2015), which forms the inspiration for Fast Multipole Attention. An $n$-body problem in physics requires evaluating all pairwise influences in a system of $n$ objects, for example, the problem of finding potentials $u_i = \sum_{j=1}^n g(\boldsymbol{x}_i, \boldsymbol{x}_j) q_j$ for $i = 1, 2, \ldots, n$ due to charges $\{q_i\}_{i=1}^n$ at locations $\{\boldsymbol{x}_i\}_{i=1}^n$, where $g(\cdot, \cdot)$ is the interaction potential of electrostatics. Attention resembles an $n$-body problem if we compare token locations to target/source locations, values to source strengths, and the attention kernel to the interaction kernel.

A popular class of methods to reduce the $\mathcal{O}(n^2)$ complexity of direct computation in $n$-body problems, the Fast Multipole Method (FMM), partitions the simulation domain into a hierarchy of cells, with each cell containing a small number of particles. It starts from the finest level and computes interactions between points in neighboring cells directly, and moves to coarser levels to compute interactions between distant source-target pairs using compact representations of points in source cells and multipole expansions. In this way, the classic tree code (Barnes & Hut, 1986) reduces the computational complexity of the $n$-body problem from $\mathcal{O}(n^2)$ to $\mathcal{O}(n \log n)$ with a hierarchical tree structure. The attention matrix of FMA has the same structure as the 1D FMM. FMMformer (Nguyen et al., 2021) approximates self-attention with a near-field and far-field decomposition using 2 levels, combining coarse linearization with local attention.

## A.10 H-MATRICES AND H-TRANSFORMER

In this section, we give some more details on the relation between the Fast Multipole Method and hierarchical matrices (H-matrices) (Hackbusch, 1999; Hackbusch & Khoromskij, 2000), which are a class of matrices with block hierarchy and are used in H-transformer (Zhu & Soricut, 2021). In an H-matrix, off-diagonal blocks are replaced by their low-rank approximation. Similar to FMA, the sparsity pattern of the attention matrix in H-transformer is hierarchical and results in linear time and memory complexity, but there are key differences. First, as a crucial advance, we use a learned downsampling method instead of the fixed averaging process used in H-transformer to compute coarse-level average quantities. Second, H-transformer's fixed averaging requires the approximation

rank $p$ to equal the group size $m$, which our experiments show is substantially sub-optimal. Third, unlike the H-matrix structure, in FMA tokens in the immediate left and right neighborhoods are not downsampled and are considered in full resolution, which results in better accuracy. Fourth, our approach has the flexibility to not downsample the queries, which leads to a better balance between accuracy and memory use in practice. Together, these key differences and advances result in FMA being substantially more accurate than H-transformer, as demonstrated in our language modeling tests.

