# OpenReview forum: "Fast Multipole Attention: A Divide-and-Conquer Attention Mechanism for Long Sequences"
_ICLR.cc/2024/Conference — Submitted to ICLR 2024_

### Official Review · Reviewer_xWY6 · 2023-10-29

**Soundness:** 2 fair
**Presentation:** 3 good
**Contribution:** 2 fair
**Rating:** 5
**Confidence:** 5

**Summary:**

This paper proposes a new efficient self-attention mechanism to reduce the $O(n^2)$ cost to $O(n)$ or $O(n \log n)$ while maintaining a global receptive filed (unlike sparse attention variants). The proposed attention uses a multi-resolution approach. It uses high resolution to approximate the  query’s attention to nearby tokens, low resolution to approximate the attention to distant tokens, and even lower resolution for even farther tokens. The authors evaluated the method on autoregressive and bidirectional language models and found better accuracy and better efficiency compared to some efficient attention baselines.

**Strengths:**

1. This method built on multi-resolution analysis, which is a powerful but less explored tool for approximation.

2. Unlike sparse attention variants, due to the use of multiple resolutions, the proposed attention provides a global receptive field to the entire sequence as well as efficiency benefits.

3. This efficient self-attention supports causal attentions. This is most useful for the recently popular large language models (LLM) since most of them are auto-regressive models.

**Weaknesses:**

1. Discussion on other efficient attentions that also uses multi-resolutions is limited. For example, MRA attention [1] also provides a multi-resolution approximation of self-attention. Also, since the proposed method is an approximation to the full self-attention matrix, it would be better to compare the approximation quality of the proposed method with other approximation baselines.

2. Error analysis in 3.6 only analyzes the error between $QK^\top$ and $Q\tilde{K}^\top$, which is too rudimentary. How does the approximation impact the final output of self-attention?

3. Experiments are only performed on 512, 1K, and 2K sequence lengths, which are not very long. Full self-attention also performs efficiently on these sequence lengths (based on my knowledge and the efficiency comparison on Figure 3). The efficiency benefits of efficient attentions are very limited on these sequence lengths. The authors should evaluate the methods on longer sequences.

4. No performance comparison to full self-attention on Table 1 and Table 2. Fused full self-attention (such as Flash Attention [2]) is integrated to PyTorch 2. The authors should give a performance and efficiency comparison to the full self-attention.

5. On efficiency comparison, the batch size is 1, so compute load is small. In this case, kernel launch latency and memory latency might account for a large portion of the overall latency. Custom CUDA kernels usually have some advantages on these latency due to fused operations. It would be more convincing to provide latency for large batch size.

[1] Multi Resolution Analysis (MRA) for Approximate Self-Attention https://arxiv.org/abs/2207.10284

[2] FlashAttention: Fast and Memory-Efficient Exact Attention with IO-Awareness https://arxiv.org/abs/2205.14135

**Questions:**

1. To the best of my knowledge, Linformer, Nystromformer, H-Transformer do not support causal attention, how are auto-regressive language modeling experiments performed on these baselines?

2. Does this method support KV cache during inference, and will it introduce more overhead? When profiling inference time for Figure 3, is KV cache used?

---

> ### Author Response · Authors · 2023-11-20
> **Response to Reviewer xWY6 (Part 1)**
>
> We thank the reviewer for the insightful and constructive comments.
>
> **Weaknesses:**
>
> 1. Yes, thanks for pointing us to MRA, which is an important contribution we were not aware about and is highly relevant. We now discuss similarities and differences with MRA and present numerical comparisons, see the response to the second referee.
>
> In short, while MRA and FMA are based on similar high-level principles, the main advance of our FMA relative to MRA attention is that our FMA approach **learns the basis functions** that compute group quantities, as opposed to the fixed wavelet family used in MRA attention.
>
> Our new numerical results show that learning the basis functions in hierarchical methods leads to substantially better accuracy for language tasks compared to MRA attention, for similar memory and time cost.
>
> Furthermore, the comment on comparing approximation quality of the attention matrix while fixing projection weights $W_Q$, $W_K$ and $W_V$ (as obtained for the pretrained full attention, for example) is interesting. However, the bottomline in our tests shows that when we train $W_Q$, $W_K$ and $W_V$ with FMA, the resulting models are more accurate than doing this with MRA or H-matrix, so *learning* the basis functions that compute group quantities in FMA leads to a large accuracy improvement compared to the results of training MRA with fixed wavelet basis. We believe this final bottomline is more representative of the desired outcome (an accurate language model with low memory cost) than direct comparison with the attention matrix as trained for a full attention model (which is too expensive to compute in practice for long enough sequence lengths).
>
> 2. More detailed error estimation as in the paper on MRA will be reported in **Part 2** of our response.
>
> 3. We agree with the reviewer that linearly scaling transformers are more advantageous for long sequence lengts, and have added results for sequence length 4096, see response to the first reviewer above. This confirms that for long sequence lengths the full transformer indeed needs very large amounts of memory.
>
> 4. We have included comparison with full attention; see the response to the first reviewer. This confirms the advantages of efficient transformers.
>
> 5. We are redoing these efficiency tests with larger batch size, to be reported in **Part 2** of our response.

---

> ### Author Response · Authors · 2023-11-22
> **Response to Reviewer xWY6 (part 2)**
>
> We provide answers for comments 1,2, and 5 below.
> 1. Also, since the proposed method is an approximation to the full self-attention matrix, it would be better to compare the approximation quality of the proposed method with other approximation baselines.
> and
> 2. Error analysis in 3.6 only analyzes the error between $QK^T$ and $Q\widetilde{K}^T$, which is too rudimentary. How does the approximation impact the final output of self-attention?
>
> **Answer to 1. and 2.:** Thank you for the suggestion. We have improved our theoretical analysis and provided the two following results on the error estimations of the attention matrix and its exponential.
>
> **Theorem 1 (Error Estimation of the Attention Matrix Approximation).**
>
> Let $\widetilde{C}$ be our two-level FMA attention matrix with $p=1$. Suppose for every $j,\ell \in \{0,\ldots,m-1\},t\in \{0,1,2,3\},$ and $i\in\{0,\ldots, n-1\}$, we have
>
> $||Q_i^T||_2\leq\beta_1$
>
> and
>
> $||K_{tm+j}^T - K_{tm+\ell}^T||_2\leq \beta_2,$
>
> and the kernel weights are normalized in the sense that  $\sum\limits_{j=0}^{m-1}\alpha_j=1$.  Then $||\widetilde{C} -C||_F$ is bounded above by
>
> $m\beta_1\beta_2\sqrt{6\sum\limits_{j=0}^{m-1}\alpha_j^2 }.$
>
> Note that the results can be easily extended to higher level $L>1$ and $p>1$.
>
> In addition, inspired by the estimation derived in  Zeng et al. (2022), we have a similar estimation of $||\exp(\widetilde{C}) -\exp(C)||_F$ where our set of components for the approximation is obtained from the hierarchical structure (see Fig. 1b).
>
> **Theorem 2.**
>
> Let $\widetilde{C}$ be the approximated attention matrix obtained from our $(L+1)$-level FMA with fixed kernel weights and $\delta_k$ be the $m_k$-th largest coefficient. Suppose
>
> $||Q_i^T||_2\leq\beta_1$
>
> and
>
> $||K_{j_1}^T - K_{j_2}^T||_2\leq \beta_2,$
>
> for $(i,j_1)$ and $(i,j_2)$ belong to the same level $I_\ell$, for $\ell = 1,\ldots L$ and $i\in\{0,\ldots, n-1\}$. Then
>
> $||\exp(\widetilde{C}) -\exp(C)||_F^2$
>
> is bounded above by
>
> $\sum\limits_{\ell=1}^LC_{2r}\delta_k^2 \left(3nm_k -6m_k^2 \right).$
>
> Here $r = 2\beta_1\beta_2$ and $C_{2r} = 1+ \exp(2r)-\exp(r)$.
>
> It is worth noting that if $\delta_k$ decays fast as $k$ increases (which we can see in fast multipole method, and can expect for 1D sequences), the error in the above estimation becomes small, illustrating the power of the fast multipole method for this kind of data.
>
> 5. Regarding this question, we use batch size 64 and include the numerical results in the updated Figure 2.

---

> > ### Comment · Reviewer_xWY6 · 2023-11-22
> >
> > Don't forget to address the Questions section

---

> ### Author Response · Authors · 2023-11-22
> **Response to Reviewer xWY6 (Part 3)**
>
> Thank you for pointing out the missing part in the response.
>
> 1. Although the authors in Nystromformer and H-Transformer do not report numerical results on autoregressive language modeling tasks, causal masking of these models can be achieved by applying a suitable causal mask. For example, for H-attention, causal masking is achieved by applying a causal mask to the blocks of the approximated attention matrix at every level, as in https://github.com/lucidrains/h-transformer-1d.
>
> In the original submission we mistakenly cited the Linformer paper, but we actually used [1].
>
> 2. We haven't implemented KV cache in our current code. Since in our experiments there is no overlap in K and V in each iteration, there is, in fact, no need to apply KV cache.
>
> [1] Transformers are RNNs: Fast Autoregressive Transformers with Linear Attention (https://arxiv.org/abs/2006.16236)

---

> ### Author Response · Authors · 2023-11-23
> **Response to Reviewer xWY6 (Part 4)**
>
> **Summary:**
>
> We believe we have now answered all the reviewers' questions.
>
> We invite the reviewer to update their score if our additional results sufficiently answer the questions by the reviewers, or let us know if further questions remain that we can address in further discussion.
>
> We thank the reviewer again for their constructive comments.

---

### Official Review · Reviewer_3Dja · 2023-10-31

**Soundness:** 2 fair
**Presentation:** 3 good
**Contribution:** 2 fair
**Rating:** 5
**Confidence:** 4

**Summary:**

This paper proposes an efficient self-attention, Fast Multipole Attention, by exploring the hierarchial structure of the input to
reduce the complexity of standard self-attention.

Fast Multiple Attention shows comparable performance to efficient Transformer variants on enwik-8 and wikitext-103 with reduced
time and memory complexity.

**Strengths:**

The motivation to propose efficient attention for reducing memory/computation cost is reasonable and the fast multipole attention scheme is interesting.

Ablation study is conducted to support the efficiency of Fast Multipole Attention.

**Weaknesses:**

One important baseline is missing, [1], which almost has the same idea that takes the multi-resolution approximation.
I would like to know what is the major difference between fast multipole attention and [1].

Since the idea is so close to [1], I would like to see how it compares with [2] on enwik-8/wikitext-103/GLUE benchmark/Long Range Arena/WikiHop.

The efficiency comparison with [1] is also needed to show the advantage of fast multipole attention.

The theoretical analysis is not as sound as [1]. What is the approximation error of fast multipole attention (see Proposition 4.5 in [1].

[1] Multi Resolution Analysis (MRA) for Approximate Self-Attention

**Questions:**

The idea is very similar to [1]. I would like to see a detailed comparison with [1]. Please see the weaknesses above. I will update my score based on the rebuttal.

---

> ### Author Response · Authors · 2023-11-20
> **Response to Reviewer 3Dja (Part 1)**
>
> We thank the reviewer for the insightful and constructive comments.
>
> In particular, we thank the reviewer for pointing out the reference to Multi-resolution analysis (MRA) attention in [1], which is indeed very relevant and is an important contribution to the field. Below we point out the similarities and differences, and compare with MRA numerically, showing that our new method has a large improvement in accuracy for the same memory cost, mainly because it **learns** the averaging operators.
>
> **Weaknesses and questions:**
>
> 1. What is the major difference between fast multipole attention and [1]?
>
> While all of H-transformer, MRA and FMA are similar at a high conceptual level (they recursively use multiple levels and resolutions to get $O(n)$ complexity while retaining the full receptive field), there are many differences in the actual mechanisms of how these methods work.
>
> However, the *seminal main advance in MRA attention from [1]* is that it **adaptively** chooses higher recursive resolution where the attention weights are the largest in the attention matrix (not necessarily close to the diagonal, which can be important, for example, for 2D data like images).
>
> H-matrix and FMA, instead, are built for 1D sequences and use higher resolution closer to the diagonal. (In fact, Multi-resolution analysis can be seen as an adaptive generalization of the Fast multipole method, see (Beylkin,1996).)
>
> The **main advance of our FMA** relative to both H-transformer and MRA attention is that our FMA approach **learns** the basis functions that compute group quantities, as opposed to the fixed averaging basis functions of H-transformer and the fixed wavelet family used in MRA attention.
>
> Our numerical results below show that **learning the basis functions in hierarchical methods leads to substantially better accuracy for language tasks compared to both H-transformer and MRA attention**, for the same memory and time cost.
>
> 2. Since the idea is so close to [1], I would like to see how it compares with [1] on enwik-8/wikitext-103/GLUE benchmark/Long Range Arena/WikiHop.
>
> The results on wikitext-103 are shown below (**see extended Table 2**):
>
> | Model | Context Size | ppl(test)↓ | Memory Footprint |
> |-------|--------------|------------|------------------|
> | MRA | 512 | 11.82 | 11.1 GB |
> | H-transformer | 512 | 10.60 | 10.7 GB |
> | FMA-m=64-3lev | 512 | **9.45** | 10.9 GB |
> | FMA-linear-m=64-3lev | 512 | 10.78 | **8.2 GB** |
> ||||
> | MRA | 1024 | 10.20 | 17.5 GB |
> | H-transformer | 1024 | 9.81 | 16.9 GB |
> | FMA-m=64-4lev | 1024 | **9.06** | 18.6 GB |
> | FMA-linear-m=64-4lev | 1024 | 10.13 | **15.6 GB** |
> ||||
> | MRA | 2048 | 9.66 | 35.3 GB |
> | H-transformer | 2048 | 9.45 | 33.1 GB |
> | FMA-m=64-5lev | 2048 | **8.95** | 37.0 GB |
> | FMA-linear-m=64-5lev | 2048 | 9.64 | **29.5 GB** |
>
> The results show that the accuracy of FMA is generally much better than H-transformer and MRA. This clearly demonstrates the advantage of **learning** the averaging operators, compared to the fixed basis functions in H-transformer and MRA.
>
> **Results for long range arena are reported in the new Table 5 in the supplement**, with discussion. Table 5 shows that FMA and FMA-linear achieve competitive results across all the tasks among the efficient transformer models, like MRA and H-matrix.
>
> As far as we know, MRA has been demonstrated for bi-directional tasks only and there are no MRA results for autoregressive language models such as used for enwik-8.
>
> 3. The efficiency comparison with [1] is also needed to show the advantage of fast multipole attention.
>
> **We have updated Figure 2** in the paper to include the efficiency comparison with [1], and find similar memory usage for MRA and FMA. This can also be seen from the table above, where FMA has better accuracy than MRA with similar memory use, due to FMA *learning* the averaging operators.

---

> ### Author Response · Authors · 2023-11-22
> **Response to Reviewer 3Dja (part 2)**
>
> 4. Our improved theoretical analysis is given below.
>
> **Theorem 1 (Error Estimation of the Attention Matrix Approximation).**
>
> Let $\widetilde{C}$ be our two-level FMA attention matrix with $p=1$. Suppose for every $j,\ell \in \{0,\ldots,m-1\},t\in \{0,1,2,3\},$ and $i\in\{0,\ldots, n-1\}$, we have
>
> $||Q_i^T||_2\leq\beta_1$
>
> and
>
> $||K_{tm+j}^T - K_{tm+\ell}^T||_2\leq \beta_2,$
>
> and the kernel weights are normalized in the sense that  $\sum\limits_{j=0}^{m-1}\alpha_j=1$.  Then $||\widetilde{C} -C||_F$ is bounded above by
>
> $m\beta_1\beta_2\sqrt{6\sum\limits_{j=0}^{m-1}\alpha_j^2 }.$
>
> Note that the results can be easily extended to higher level $L>1$ and $p>1$.
>
> In addition, inspired by the estimation derived in  Zeng et al. (2022) (MRA), we have a similar estimation of $||\exp(\widetilde{C}) -\exp(C)||_F$ where our set of components for the approximation is obtained from the hierarchical structure (see Fig. 1b).
>
> **Theorem 2.**
>
> Let $\widetilde{C}$ be the approximated attention matrix obtained from our $(L+1)$-level FMA with fixed kernel weights and $\delta_k$ be the $m_k$-th largest coefficient. Suppose
>
> $||Q_i^T||_2\leq\beta_1$
>
> and
>
> $||K_{j_1}^T - K_{j_2}^T||_2\leq \beta_2,$
>
> for $(i,j_1)$ and $(i,j_2)$ belong to the same level $I_\ell$, for $\ell = 1,\ldots L$ and $i\in\{0,\ldots, n-1\}$. Then
>
> $||\exp(\widetilde{C}) -\exp(C)||_F^2$
>
> is bounded above by
>
> $\sum\limits_{\ell=1}^LC_{2r}\delta_k^2 \left(3nm_k -6m_k^2 \right).$
>
> Here $r = 2\beta_1\beta_2$ and $C_{2r} = 1+ \exp(2r)-\exp(r)$.
>
> It is worth noting that if $\delta_k$ decays fast as $k$ increases (which we can see in fast multipole method, and can expect for 1D sequences), the error in the above estimation becomes small, illustrating the power of the fast multipole method for this kind of data.

---

> ### Author Response · Authors · 2023-11-23
> **Response to Reviewer 3Dja (Part 3)**
>
> **Summary:**
>
> We believe we have now answered all the reviewers' questions.
>
> We invite the reviewer to update their score if our additional results sufficiently answer the questions by the reviewers, or let us know if further questions remain that we can address in further discussion.
>
> We thank the reviewer again for their constructive comments.

---

### Official Review · Reviewer_AtqH · 2023-11-06

**Soundness:** 3 good
**Presentation:** 3 good
**Contribution:** 3 good
**Rating:** 5
**Confidence:** 4

**Summary:**

This paper proposes an attention mechanism with $O(N\log(N))$ complexity where N is the sequence length. In particular, the paper proposes that each query attends to a local neighborhood in full resolution and to points further in the sequence at exponentially lower resolutions. The low resolution tokens are computed using strided convolution. The experiments in the paper compare the described method on autoregressive language modeling on enwik8 and masked language modeling on Wikitext-103, where it outperforms several other efficient attention methods.

**Strengths:**

The proposed method is conceptually simple and intuitive. In addition, the authors perform controlled testing, namely same exact model with different attention implementations, on real world tasks, autoregressive and masked language modeling. Finally,  the paper is well written with a sufficiently comprehensive review of the related works.

**Weaknesses:**

The major issue with the paper has to do with the experimental evaluation. Although I appreciate using real world language modeling tasks, there is a major baseline lacking, namely the vanilla transformer. This becomes even more important given that the setup is unusually small with a 6 layer transformer only. For instance, Reformer reports results with 12 layers on enwik8 which are significantly better at 1.19 bits per dimension. How would vanilla transformer and FMA compare at those sizes?

Another quite significant omission is the lack of any ablations. The method has several hyper parameters that change from experiment to experiment. There is absolutely no ablation study to provide some insight on the effect of group size m and the number of levels to the performance as well as the FLOPS and memory required.

**Questions:**

- How does vanilla transformer perform in the provided experiments?
- How does memory compressed transformer perform in the same experiments? Dividing the computational and memory cost by 3 would be quite enough for the tested sequence lengths.
- Minor, but why would the causal linear attention require more memory than the non-causal version? This seems like a bug in the implementation.

---

> ### Author Response · Authors · 2023-11-20
> **Response to Reviewer AtqH (Part 1)**
>
> We thank the reviewer for the insightful and constructive comments.
>
> **Weaknesses and questions:**
>
> We agree with the reviewer that more extensive experimental evaluation is warranted.
>
> We have added the requested results in extended tables in the paper:
> * full comparison with the vanilla transformer: **see extended Tables 1 and 2 (additions in blue), and updated Figure 2**
>
> **It can be seen that full attention, as expected, has the best
> accuracy but its memory consumption grows quadratically and becomes impractical for large sequence lengths.** Efficient transformers aim to obtain linear scaling in memory use while retaining good accuracy (so the models can use longer sequences).
>
> * comparison with the memory compressed transformer: **see extended Table 1 (additions in blue)**
>
> **Compressed attention reduces the memory by a factor of 3 but loses precision and still scales quadratically in memory; for large sequence lengths, it has worse accuracy and larger memory use than FMA.**
>
> * an ablation study on the effect of group size m (and number of levels) and rank p: **see new Table 3 in supplement**
>
> It can be seen, as expected, that bpc improves as group size m grows at the cost of larger memory use.
> Similarly, bpc improves, as expected, as rank p grows at the cost of larger memory use.
>
> FMA is quite flexible in allowing the user to tune m and p.
>
> The choices for m and p in the experiments were suitable for the problem types and sizes we considered, compared to competing methods.

---

> ### Author Response · Authors · 2023-11-22
> **Response to Reviewer AtqH (Part 2)**
>
> * Results with 12 layers: **see new Table 4 in the supplement**
>
> Note that for full attention, we had to reduce the batch size from 32 to 16 and accumulate the gradient over batches to avoid GPU memory overflow, as listed in the accumulated memory use. Note that, unfortunately, we don't have access to multiple GPUs that we can use in parallel, so given the time constraints we were only able to do a few tests with 12 layers. These test results show similar trends as our findings in the other tables.
>
> **Question on memory use of linear attention:**
>
> Regarding the question about the memory of the causal linear attention and the non-causal one, we use the codes from https://github.com/lucidrains/linear-attention-transformer
> and report the memory. There could be a bug in the implementation. We will investigate this issue in the future.
>
> **Summary:**
>
> We believe we have now answered all the reviewers' questions.
>
> We invite the reviewer to update their score if our additional results sufficiently answer the questions by the reviewers, or let us know if further questions remain that we can address in further discussion.
>
> We thank the reviewer again for their constructive comments.

---

### Author Response · Authors · 2023-11-23
**Summary of rebuttal and changes:**

We have now uploaded the final rebuttal version of the paper (Wed Nov 22, 12:30pm EST). (All changes are indicated in blue color.)

We believe we have comprehensively answered all questions by the three reviewers.

Our additional results show conclusively that **our new Fast Multipole Attention (FMA) method decisively improves accuracy for language tasks relative to the state of the art in hierarchical efficient transformers: Multi-Resolution Analysis attention (MRA, Zeng et al., ICML 2022)  and H-transformer (Zhu & Soricut, IJC-NLP 2021).**

Our FMA transformer is built on similar high-level ideas as H-transformer and MRA attention, but the actual methodology has substantial differences.

Our **key advance** relative to MRA attention and H-transformer is that we **learn the basis functions** that compute group quantities on all recursive levels, as apposed to the fixed basis functions in MRA and H-transformer. While we do not have the adaptivity of MRA, our results show that **learning the basis functions improves accuracy dramatically.** An advantage of FMA is that it has a relatively simple recursive structure (compared to MRA), suitable for 1D data where long-range dependencies are weaker than close-by dependencies.

We have also performed all other additional tests that were requested and have improved the theoretical error estimation (inspired by MRA (Zeng et al., ICML 2022)) as requested by the reviewers.

We thank the reviewers again for their constructive comments.

We now invite the reviewers to update their scores if our additional results sufficiently answer the questions by the reviewers, or let us know if further questions remain that we can address in further discussion.

We added the code to the supplementary material

---

### Meta-Review · Area_Chair_beZL · 2023-12-13

**Metareview:**

All reviewers find the proposed approach interesting and the initial results promising. However reviewers find the paper still in early stages and the main concern is about the limited experiments presented in the paper. Authors have provided some more additional experiments, but more wider settings/longer context experiments are required to understand the method better. Adding latency results also can help the paper. I encourage authors to continue building on this work and make a stronger submission in the future, but have to reject the current submission.

**Justification For Why Not Higher Score:**

limited experiments

**Justification For Why Not Lower Score:**

N/A

---

### Decision · Program_Chairs · 2024-01-16

Reject